# Myricetin Amorphous Solid Dispersions—Antineurodegenerative Potential

**DOI:** 10.3390/molecules29061287

**Published:** 2024-03-14

**Authors:** Natalia Rosiak, Ewa Tykarska, Judyta Cielecka-Piontek

**Affiliations:** 1Department of Pharmacognosy and Biomaterials, Faculty of Pharmacy, Poznan University of Medical Sciences, 3 Rokietnicka St., 60-806 Poznan, Poland; nrosiak@ump.edu.pl; 2Department of Chemical Technology of Drugs, Poznan University of Medical Sciences, 3 Rokietnicka St., 60-806 Poznan, Poland; etykarsk@ump.edu.pl

**Keywords:** amorphous solid dispersion, myricetin, glass transition, molecular docking, miscibility, DFT calculation, neuroprotection

## Abstract

Our research aimed to develop an amorphous solid dispersion (ASD) of myricetin (MYR) with Polyvinylpyrrolidone K30 (PVP30) to enhance its solubility, dissolution rate, antioxidant, and neuroprotective properties. Employing a combination of solvent evaporation and freeze drying, we successfully formed MYR ASDs. XRPD analysis confirmed complete amorphization in 1:8 and 1:9 MYR-PVP weight ratios. DSC thermograms exhibited a single glass transition (Tg), indicating full miscibility. FT-IR results and molecular modeling confirmed hydrogen bonds stabilizing MYR’s amorphous state. HPLC analysis indicated the absence of degradation products, ensuring safe MYR delivery systems. Solubility, dissolution rate (pH 1.2 and 6.8), antioxidant (ABTS, DPPH, CUPRAC, and FRAP assays), and in vitro neuroprotective activities (inhibition of cholinesterases: AChE and BChE) were significantly improved compared to the pure compound. Molecular docking studies revealed that MYR had made several hydrogen, hydrophobic, and π-π stacking interactions, which could explain the compound’s potency to inhibit AChE and BChE. MYR-PVP 1:9 *w*/*w* ASD has the best solubility, antioxidant, and neuroprotective activity. Stability studies confirmed the physical stability of MYR-PVP 1:9 *w*/*w* ASD immediately after dissolution and for two months under ambient conditions. Our study showed that the obtained ASDs are promising systems for the delivery of MYR with the potential for use in alleviating the symptoms of neurodegenerative diseases.

## 1. Introduction

Polyphenols constitute a class of bioactive compounds found in various fruits, vegetables, and other plant-based sources. Widely studied for their potential health benefits, which include antioxidant, anti-inflammatory, and anticancer properties, polyphenols play a crucial role in neuroprotection by serving as effective antioxidants that mitigate oxidative stress [1,2,3,4,5]. Research has shown that diets rich in polyphenols or supplemented with specific polyphenols can significantly reduce oxidative damage in the brain, thereby preserving both neuronal structure and function. This becomes particularly relevant in the context of neurodegenerative diseases, where oxidative stress is a prominent feature. Polyphenols have been shown to inhibit the formation of beta-amyloid plaques and neurofibrillary tangles in Alzheimer’s disease, as well as prevent the aggregation of alpha-synuclein in Parkinson’s disease [6,7,8,9,10].

However, one of the challenges associated with the therapeutic application of polyphenols is their limited water solubility, which hampers their bioavailability and subsequent efficacy. One such polyphenol is myricetin (3,5,7,3′,4′,5′-hexahydroxyflavone, MYR). The presence of MYR was confirmed in numerous plant families, such as the *Myricaceae*, *Polygonaceae*, *Primulaceae*, *Pinaceae*, and *Anacardiaceae* [11]. In addition, MYR is primarily found in berries, fruits, vegetables, teas, and wine. MYR’s antioxidant and nutraceutical benefits are well supported by the literature [12]. For example, MYR exhibits anti-inflammatory [13], neuroprotective [14], cardiovascular [15], antidiabetic [16], hepatoprotective [17], antitumor [18] effects, antioxidant properties, and free-radical-scavenging effects [19]. MYR has been the focus of research in many clinical trials, both alone and in combination with other flavonoids. The literature reports preclinical and clinical studies for the treatment of diabetes [20,21] and chemopreventive abilities [22].

Therefore, it is essential to use efficient techniques, such as delivery systems, to improve the solubility, absorption, and therapeutic potential of MYR. In the literature, attempts to improve the solubility of MYR using various techniques have been reported (e.g., cocrystals [23,24], micelles [25,26,27], inclusion complex [28,29], nanofibers [30], nanoliposomes [31], solid dispersion [32,33], amorphous solid dispersion [32,34], tooth-binding nanoparticles [35], nanoliposomes [36], lipid nanoparticles [37], pro-liposomes [38], and nanophytosomes [39]). For example, micelles obtained by Wang et al. [25] significantly improved the solubility and effectiveness of MYR against glioblastoma cells. Micelles demonstrated enhanced drug-release profiles, increased glioblastoma cell cytotoxicity, and improved brain affinity. In another study [29], the MYR/HP-β-CD inclusion complex exhibited enhanced water solubility, dissolution rate, and oral bioavailability (9.4-fold increase) in rats compared to free myricetin. Crystalline solid dispersion of MYR obtained by Huang et al. [33] enhanced MYR dissolution but scientists did not check the biological activity of the obtained dispersion. Mureşan-Pop et al. [32] and Zhang et al. [34] obtained amorphous solid dispersions (ASDs). Mureşan-Pop et al. [32] confirmed improved solubility in deionized water (pH 5.8). They also did not conduct any tests on the biological activity of the obtained ASDs, whereas Zhang et al. [34] confirmed that obtained ASDs have higher equilibrium solubility than pure MYR and can scavenge active oxygen free radicals and provide antioxidative effects in biological systems (based on cell viability characterization and ROS-level detection experiments).

The literature reports that amorphous solid dispersions (ASDs) are becoming one of the leading techniques to improve the solubility of polyphenols. ASDs refer to the preparation technique where a poorly water-soluble drug or compound, such as polyphenol, is dispersed in a hydrophilic carrier matrix [40]. The use of ASD technology allows for the creation of a stable, amorphous structure that enhances the dissolution properties of polyphenols, thereby increasing their bioavailability. Over the past years, the preparation of ASDs of curcumin [41,42,43], daidzein [44], genistein [45], hesperidin [46,47], hesperetin [47,48], myricetin [32,34], naringenin [49], pterostilbene [50], quercetin [51,52,53], resveratrol [42], and sinapic acid [54] has been confirmed. These works confirm that the amorphization process makes it possible to significantly improve the solubility and/or biological activity of polyphenols.

Prior works on obtaining solid dispersions and amorphous solids dispersions of myricetin, thus far, have not considered their impact on biological properties [32,33,34]. Our research represents the first comprehensive study, providing not only a broad physicochemical characterization but also delivering insights into the antioxidant and neuroprotective properties of the obtained ASDs through in vitro testing.

## 2. Results and Discussion

Polyvinylpyrrolidone (PVP, Figure 1a) is commonly used in the formulation of amorphous solid dispersions (ASDs) to enhance the solubility and biological activity of poorly soluble polyphenols. PVP acts as a hydrophilic polymer that can form a stable molecular dispersion with polyphenols, thereby preventing their crystallization and maintaining them in an amorphous state. It was confirmed that the amorphous form of the polyphenols has a significantly higher solubility compared to its crystalline counterpart [45,46,48,55,56,57,58]. Therefore, this study attempts to obtain an amorphous solid dispersion (ASD) of myricetin (MYR, Figure 1b) with PVP K30 (PVP30).

The first stage of the study concerned the preparation of ASD of MYR and PVP30 by a combination of solvent evaporation and freeze-drying techniques. Next, MYR, PVP30, and the ASD thereof were characterized by X-ray powder diffraction (XRPD), thermogravimetric analysis (TG), differential scanning calorimetry (DSC), and Fourier transform infrared (FT-IR). Then, the properties of amorphous MYR in ASD (apparent solubility, antioxidant, and neuroprotective activity) were investigated.

X-ray powder diffraction (XRPD) analysis is a commonly used technique in material science to determine the crystalline or amorphous nature of a sample. Amorphous materials lack long-range order in their atomic arrangement, while crystalline materials exhibit a regular and repeating pattern. Confirming the amorphous/crystalline nature of a substance is crucial as it affects its physical and chemical properties, including solubility, stability, and bioavailability [46,47,48,50,55,56,59].

In this study, XRPD analysis was employed to confirm the nature of MYR in physical mixtures (ph. m.) and in ASDs. The powder diffraction patterns of pure MYR showed characteristic high-intensity Bragg peaks and confirmed crystalline form (Figure 2, black line). The diffractogram of PVP30 (Figure 2, red line) displayed no sharp peaks due to its amorphous nature. The literature confirms that the amorphous form of PVP30 is characterized by the presence of two broad peaks with a maximum at around 12° 2Θ and 20° 2Θ [60].

Diffractograms of pure compounds were then compared to the diffractograms of physical mixtures and ASDs. The diffractograms of MYR-PVP 1:8 and 1:9 (*w*/*w*) physical mixtures showed the sum of the MYR and PVP30 peaks (Figure 2, light green and light blue, respectively). Our previous study confirmed that polyphenol-polymer physical mixtures show characteristic peaks of polyphenol with reduced intensity [46,48,50]. The disappearance of the Bragg peaks in the diffractograms of 1:8 *w*/*w* and 1:9 *w*/*w* MYR-PVP ASDs (Figure 2, green and blue, respectively) is an indication sign of the loss of long-range order typically associated with crystalline structures. The absence of well-defined peaks (effect “halo”) suggests that MYR exists in an amorphous form within the PVP30 matrix. This finding aligns with previous research highlighting the ability of PVP to promote the amorphous state of polyphenol molecules [44,45,61,62]. Based on XRPD analysis, Mureşan-Pop et al. [32] indicated that the solid dispersions of MYR obtained by spray drying with PVP showed peaks of weak intensity, confirming incomplete amorphization of MYR. Therefore, the combination of solvent evaporation and freeze-drying methods seems to be a more efficient method for MYR amorphization than the spray-drying method.

The differential scanning calorimetry (DSC) provides information on melting, crystallization, decomposition processes, or changes in heat capacity. It is very useful for learning about the thermodynamic properties of a solid. In our study, DSC analysis was used to confirm the amorphization of MYR in ASDs with PVP30 and the miscibility of the obtained ASDs. The MYR thermogram revealed a sharp peak at 364 °C, which indicates melting of crystalline MYR (Figure 3 and Appendix A, black line). This is consistent with literature data that state that MYR has a melting point of approximately 368 °C [63]. As shown by the TG thermogram (Appendix A, blue line, Appendix A), there was a 40% mass loss of MYR at the melting temperature signifying the decomposition of polyphenol. 

The DSC thermograms of MYR-PVP 1:8 and 1:9 (*w*/*w*) ASDs (Figure 3) displayed the absence of a melting endotherm, indicating that MYR no longer exists in a crystalline state. The conversion from a crystalline to an amorphous state can be attributed to the strong interactions between MYR and PVP30, preventing the recrystallization of the MYR during the preparation process. This is consistent with the results obtained from the XRPD analysis, which supports the conclusion that MYR exists in an amorphous state in obtained ASDs.

A significant observation in the DSC thermograms was the presence of a single glass transition temperature (T_g_) at about 178.4 °C and 179.7 °C in ASDs, in the ratio 1:8 *w*/*w* and 1:9 *w*/*w*, respectively. The glass transition is a characteristic event associated with the transition from a “rigid” amorphous state to a more flexible, “rubbery” state. The appearance of a distinct glass transition temperature in the thermogram suggests the formation of a homogenous ASD. This finding indicates that the polyphenol and PVP have a strong molecular interaction, resulting in the formation of a single, molecular-level glassy phase.

The presence of a single T_g_ is of great importance in ASD. It suggests that the drug and polymer are molecularly dispersed and do not undergo phase separation during the preparation or storage. Phase separation can lead to the formation of drug-rich regions, which may result in poor drug-release kinetics and reduced bioavailability. The absence of multiple T_g_ peaks in the DSC thermogram confirms the formation of a stable and homogeneous amorphous ASD, which may affect the potential for improved drug dissolution and absorption.

Research studies provide that the lack of melting effect and the appearance of T_g_ confirm full amorphization [48,50,64]. For example, amorphous polyphenol-PVP dispersions for which the melting point of polyphenol disappeared were presented for curcumin [62], daidzein [44], genistein [45], naringenin + hesperetin [56], and wogonin [65]. Additionally, researchers show that a single glass transition seen in an amorphous dispersion’s thermogram indicates complete miscibility [50,66,67]. For instance, based on thermal analysis, the full miscibility of curcumin-piperine-PVP VA64 [55], genistein-amino acids [68], hesperidin-Soluplus^®^ and hesperidin-HPMC [46], hesperetin-piperine-PVPVA64 [48], kaempferol-Eudragit^®^ [46], pterostilbene-Soluplus^®^ [50], and tannic acid-poly(ε-caprolactone) [69] were confirmed.

FT-IR allows the identification of functional groups and molecular interactions in amorphous solid dispersions. It provides information about the presence of specific bonds and interactions between the polymer and other components in the dispersion. This analysis helps assess the drug–polymer compatibility and potential interactions that may influence the stability and performance of the ASD formulation [46,50,54,59,70]. In this study, infrared spectroscopy was used to demonstrate potential interactions between MYR and PVP30. The FT-IR spectra of pure MYR and PVP30 were compared with MYR-PVP 1:8 and 1:9 *w*/*w* physical mixtures and ASDs (Figure 4a,b).

The most characteristic absorbance peaks of MYR are located at about 1020 cm^–1^ (C–O stretching vibrations), 1159 cm^–1^ (C–O–C vibrations), 1271 cm^–1^ (C–O stretching and C–OH bending) 1323 cm^–1^ (O–H in-plane bending vibrations or C–C stretching of B ring), 1375 cm^–1^ (C–C stretching), 1512 cm^–1^ (C=C stretching of A ring), 1591 cm^–1^ (C=O stretching vibration), 1659 cm^–1^ (C=O stretching vibration) (Figure 4a). Between 2800 cm^–1^ and 3600 cm^–1^ (Figure 4b) observed peaks at about 3264 cm^–1^ (C–H stretching vibration) and 3404 cm^–1^ (O–H stretching vibration) corresponding to the hydrogen bonding.

In the FT-IR spectra of PVP, peaks can be distinguished at about 1229 cm^−1^ (lactone structure), 1269 cm^−1^ and 1285 cm^−1^ (C–N stretching), 1373 cm^−1^ (CH_2_ bending), 1420 cm^−1^ (C–H), 1458 cm^−1^ (O–H bending), 1665 cm^−1^ (–C=O stretching of tertiary amide in N-vinylpirollidon), 2876 cm^−1^ and 2951 cm^−1^ (C–H stretching) [6,15,62,65,71].

The FT-IR spectra of physical mixtures showed peaks, that were the sum of peaks characteristic of MYR and PVP30. This can simply be considered a superposition of the two components. After amorphization, the functional groups in the FT-IR spectra of MYR-PVP ASDs show band shifts, broadening, and disappearance compared to the spectra of pure MYR and PVP30 (Figure 4, dark blue and dark green lines).

The complete disappearance of MYR peaks in the spectrum of the ASDs in the 400–1750 cm^−1^ range confirms the full dispersion of the compound in the polymer matrix. The main changes were observed for peaks of PVP30. For example, the carbonyl band (PVP30, 1665 cm^−1^) is shifted to lower wavenumbers. Muresan-Pop et al. [32] show similar observations and suggest the formation of a hydrogen bond between the carbonyl group of PVP30 and the phenol O–H group in MYR. In another study, He et al. suggested that the carbonyl group of pyrrolidone in PVP30 was a proton acceptor and potentially could form hydrogen-bonding with curcumin [62]. In addition, the O–H absorption band observed at about 2800–3600 cm^−1^ that is characteristic of MYR is broad and shifted in the spectra of the ASDs (Figure 4b), which also indicates the trapping of MYR inside the PVP30 matrix.

All results suggested that intermolecular O–H⋯O=C bonds were formed between MYR and PVP30. This is in line with the literature reports [61,72,73]. Sathishkumar et al. [73] also observed a band shift in the range of 3420 cm^−1^ for myricetin-mediated gold nanoparticles. They indicated that the hydroxyl groups of MYR are involved in interactions. According to Nadal et al. [61], PVP’s chemical structure suggests that it can behave as a proton acceptor because of the carbonyl oxygen (peak measured at around 1665 cm^−1^) or nitrogen in the pyrrole ring (peaks observed at approximately 1269 cm^−1^ and 1285 cm^−1^). The carbonyl group is more conducive to hydrogen bonding because the steric hindrance effect prevents the nitrogen from taking part in intermolecular interactions.

Molecular modeling was used to complement the FT-IR analysis. Our previous study confirmed that this technique helps in the comprehensive understanding of interactions taking place in ASD of polyphenol-polymer. By AutoDock 4.0 software, the possible rotations of MYR’s bonds and interactions between MYR and PVP30 (Figure 5b) were determined.

MYR has seven active rotatable bonds between atoms O2:C13, O3:C15, O5:C19, O6:C21, O7:C22, O8:C23, C10:C12 (Figure 5a). A simple theoretical MYR-PVP model showed that the O-H group of MYR and the C=O group of PVP30 form hydrogen bonds. The confirmation of the same interaction between MYR and PVP in both the FT-IR study and molecular docking analysis strengthens the validity and reliability of the findings. The FT-IR study provides experimental evidence of the intermolecular hydrogen interactions between MYR and PVP, while the molecular docking study complements these findings by providing a theoretical understanding of the binding mechanisms. This consistency between the experimental and theoretical approaches adds robustness to the conclusion that MYR forms strong interactions with PVP in the ASDs. Such strong hydrogen interactions are crucial for the stability and performance of the dispersion, as they ensure the effective incorporation and dissolution of MYR within the PVP matrix (confirmed in XRPD and DSC study).

Compared to their crystalline counterparts, amorphous polyphenols frequently have better solubility and dissolution rates. Improved bioavailability and therapeutic effectiveness may result from this increased solubility [46,47,48,50,55].

In our investigation, we explored the impact of MYR amorphization on its apparent solubility. The results of the HPLC analysis affirmed that the amorphization process did not lead to the chemical degradation of the compound. The apparent solubility of the pure MYR was far below 0.30 ± 0.02 µg/mL, categorizing it as practically insoluble [63]. MYR-PVP ASDs led to an increase in apparent solubility. Amorphization of MYR improved the apparent solubility by ~1283-fold and ~1426-fold for MYR-PVP 1:8 *w*/*w* (385 ± 3 µg/mL) and 1:9 *w*/*w* (428 ± 2 µg/mL) ASDs, respectively. Prior research supports the notion that amorphous solid dispersions with PVP contribute to improved solubility of poorly soluble drugs and polyphenols, as evidenced in various studies [44,45,55,56,62,74,75,76,77]. Examples including amorphous solid dispersions of resveratrol-PVPK29/32 [77], curcumin-PVPK29/32 [76], curcumin-piperine-PVPVA64 [55], hesperetin-piperine-PVPVA64 [48], naringenin-hesperetin-PVP30 [56], genistein-PVP30 [45], daidzein-PVPK90 [44], and felodipine-PVP30 [75] were confirmed.

After testing the solubility of MYR-PVP 1:9 *w*/*w*, the aqueous solution was dried to assess the physical state of MYR. XRPD diffractograms conducted after the solubility analysis confirmed the amorphous nature of MYR (see Figure 6).

Changes in the 2Θ angles and the intensity of the two characteristic maxima of MYR-PVP ASD can be related to water sorption by sample. Teng et al. [78] report that there is a significant correlation between the water content in PVP samples exposed to moisture and the observed shifts in the position of the halo effect in the XRPD patterns.

The dissolution profiles of MYR-PVP 1:9 *w*/*w* were analyzed in solutions of pH 1.2 and pH 6.8 corresponding to the gastric and the intestinal environments, respectively (Figure 7). ASD showed noticeable improvement in release rate in these fluids.

The dissolution profile of MYR-PVP 1:9 *w*/*w* in a solution with pH 1.2 in the initial time of 15 min exhibited a dissolution rate of 38.70% ± 0.14%. As the dissolution process progressed, the dissolution rates increased to 43.11% ± 0.04% at 30 min and maintained a plateau state until the end of the test (6 h). According to the literature [79,80], the first 1–2 h are crucial for the gastric pH dissolution test. Also, Zhang et al. [34] for ASD of MYR-PVP observed a plateau at pH 1.2. However, the solubility maximum (~75%) was reached after about 10 minutes.

In a pH 6.8, the dissolution profile of MYR was greater than in acidic conditions, according to the weakly acidic nature of MYR (pK_a_ of 6.63 ± 0.09) [81]. In addition, the “spring and parachute” effect was observed. This phenomenon involves the creation of a rapidly dissolving and supersaturated “spring” that prevents precipitation and acts as a “parachute” that maintains the supersaturated state [82]. PVP served as a “spring” to achieve peak supersaturation of MYR by forming amorphous MYR via intermolecular force (hydrogen bonds formed between MYR and PVP), which was confirmed by FT-IR analysis. Eventually, the MYR concentration decreased to ~60%, suggesting recrystallization of ASD. Nevertheless, MYR-PVP 1:9 *w*/*w* guarantees the preservation of the supersaturation condition for up six hours, which is approximately as long as intestinal contents can remain in the duodenum, the body’s primary site of absorption.

For the same combined methods used to produce MYR amorphous solid dispersions (ASDs), Zhang et al. [34] obtained a higher solubility of MYR at pH 1.2 (they did not perform tests at pH 6.8). Differences in obtained results may be from the solvents used. In our study, ASD of MYR was prepared in the first step in ethanol and the second step in water. Zhang et al. [34] utilized a mixture of ethanol (EtOH) and dichloromethane (DCM) in the 1:2 *v/v* ratio in their study. It is noteworthy that DCM has been classified as a probable human carcinogen based on evidence of carcinogenicity in mice [83]. Also, the literature indicates that DCM can contribute to several types of cancer (liver and kidney cancer, lymphoma, and leukemia) [84]. In addition, DCM significantly contributes to smog formation and environmental pollution [84]. Meanwhile, the FDA labeled ethanol as a generally recognized as safe (GRAS) substance. Furthermore, Le Dare et al. [85] published a review work in which they present the therapeutic applications of ethanol and its efficacy and safety profiles across various conditions. Due to this, the approach used in the presented work represents a safer alternative to prepare ASD of MYR.

The literature reports that improving apparent solubility can positively affect the biological properties of the compound [28,30,46,47,48,50,53,54,55,56,62]. Therefore, we examined how improving the solubility of MYR introduced into amorphous solid dispersions translated into changes in its biological properties, such as antioxidant potential and inhibition of enzymes influencing the development of neurodegenerative diseases.

MYR exhibits the scavenging activity towards a number of radicals and ions [86,87]. A lot of reports have been published concerning the antioxidant activity of MYR, leaving no doubt that the compound is a powerful antioxidant [86,88,89,90,91]. The impact of MYR amorphization on antioxidant activity was studied using four tests (ABTS—2,2’-azino-bis(3-ethylbenzothiazoline-6-sulfonic acid) radical cation-based assays, DPPH—2,2-diphenyl-1-picrylhydrazyl assay, CUPRAC—cupric reducing antioxidant capacity assay, FRAP—ferric reducing ability of plasma assay). Due to its very low solubility, the crystalline form of MYR exhibited no activity. On the other hand, the antioxidant properties were greatly improved by MYR-PVP ASDs (see Table 1). In every test assessing antioxidant activity, MYR-PVP 1:9 *w*/*w* ASD consistently demonstrated superior efficacy.

Comparing the obtained data with the literature, it can be indicated that MYR-PVP 1:9 *w*/*w* showed stronger antioxidant activity in the ABTS assay than nanofiber MYR [30], MYR micelles [27], and in the DPPH assay than nanoencapsulated MYR [91]. Water solution of MYR-PVP 1:9 *w*/*w* had a lower IC_50_ value in the ABTS test when compared to nanofibers MYR in water (ASD IC_50_: 9.53 ± 0.17 µg/mL, nanofibers IC_50_: 11.31 ± 0.44). Myr micelles obtained by Hou et al. [27] achieved an ABTS scavenging level of only 22.20% and 41.77% for 15.6 μg/mL MYR after 15 and 120 min of incubation. The free radical DPPH scavenging was 50.0% at ~0.1 µM MYR-PVP, wherein nanoencapsulated MYR at low dose (50 uM) and high dose (250 uM) have activity of ~39.2% and ~54.6%, respectively. MYR-PVP 1:9 *w*/*w* showed higher activity in the FRAP assay than MYR micelles [26] and lower activity than MYR nanofibers [30]. To date, CUPRAC studies have not been conducted for MYR delivery systems [90].

The literature reported that MYR has the capability to inhibit acetylcholinesterase (AChE) and butyrylocholinoesterase (BChE) enzymes [92,93]. Increasing MYR’s solubility could result in an expansion of its bioavailability and help it reach an effective concentration where it can interact with the active center of cholinesterases that are essential for the onset of neurodegeneration. The inhibitory activities of water solutions of MYR and MYR-PVP ASDs on AChE and BChE were evaluated in vitro. Also, in this case, due to its practical insolubility, the crystalline form of MYR exhibited no activity. On the other hand, improved solubility had a positive effect on neuroprotective activity. MYR-PVP ASDs inhibited AChE with IC_50_ values of 23.31 ± 1.03 µg/mL and 26.12 ± 1.06 µg/mL and BChE with IC_50_ values of 87.54 ± 0.92 µg/mL and 90.5 ± 1.12 µg/mL (1:8 *w*/*w* and 1:9 *w*/*w* respectively). MYR-PVP ASDs seem more selective towards BChE than to AChE due to their higher potency towards BChE than AChE. The selective index for BChE (SI_(AChE/BChE)_) was 3.76 and 3.46 for MYR-PVP 1:8 *w*/*w* and 1:9 *w*/*w* respectively). The selectivity of MYR can be attributed to differences in the active sites of AChE and BChE enzymes [94,95]. BChE has a larger and more flexible active site compared to AChE, allowing for a wider range of interactions with inhibitors such as MYR. This structural difference may contribute to the higher affinity of myricetin towards BChE.

Molecular docking (MD) is a key technique in structural molecular biology, as confirmed by numerous literature reports [96,97,98,99,100]. MD was used to observe and confirm possible interactions between MYR and AChE/BChE, which are important targets for neuroprotection.

The neuroprotective activity of the MYR-PVP ASDs, as demonstrated by the AChE and BChE tests, indicates a potential role in supporting cognitive function and protecting against neurodegenerative conditions. The molecular docking experiments were carried out in the active site 3D space of both the AChE and BChE, using the Autodock Tool program, to gain insight into the intermolecular interactions. Active site gorges of AChE (PDB id: 4BDT) and BChE (PDB id: 4BDS) were shown in Figure 8a,c.

Docking of MYR to AChE (binding energy: −11.86 kcal·mol^−1^) and BChE (binding energy: −6.75 kcal·mol^−1^) it revealed conspicuous hydrophobic interactions, hydrogen bonds, and π-stacking interactions (Figure 8b and Figure 8d, respectively). The lowest energy conformer of MYR showed hydrogen bonds with SER^179^, GLY^116^, TYR^129^, TYR^337,^ and THR^79^ and hydrophobic interaction with TRP^435^. MYR located parallel to TRP^72^ and TYR^333^ constitute π-π stacking.

Due to a difference in the amino acid chain in the active site of BChE, the kind of binding interactions between MYR molecule and BChE was different than the interactions between MYR molecule and AChe. At the active site of BChE, MYR showed hydrogen bonds with GLY^112^ and TYR^125^, hydrophobic interactions with PHE^326^, PHE3^93^, and TRP^79^, and π-stacking interaction with HIS^433^ (see Figure 8d).

The results obtained are in line with the literature. For example, Yener et al., on the basis of their experimental and theoretical studies, indicate myricetin as a compound showing potential inhibitory effects on AChE and BChE enzymes. In addition, they confirmed that the molecular modeling data they obtained matched well with the results of the in vitro studies conducted [101].

The stability study was conducted over a period of zero, one, two, and three months for MYR-PVP 1:9 *w*/*w* ASD. After one and two months, no observable changes were noted. After three months, a notable change in the XRPD spectrum character was observed (Appendix A, Appendix A). An additional intense peak was recorded with a maximum at around 14.4° 2Θ. These indicate the onset of recrystallization in the ASD.

Our research highlights the potential of amorphous solid dispersions as a promising approach for improving the bioavailability and therapeutic efficacy of MYR.

## 3. Materials and Methods

### 3.1. Materials

Myricetin was purchased from Xi’an Tian Guangyuan Biotech Co., Ltd., Xi’an, Shaaxi Province, China. PVP30 was supplied by BASF SE (Ludwigshafen am Rhein, Germany). Acetonitrile (high-performance liquid chromatography (HPLC) grade) and formic acid (85%) were provided by POCH (Gliwice, Poland). High-quality pure water was prepared using a Direct-Q 3 UV purification system (Millipore, Molsheim, France, model Exil SA 67120). Methanol was supplied by Chempur (Piekary Śląskie, Poland).

### 3.2. Preparation of Amorphous Solid Dispersion of Myricetin

Based on the screening study we confirmed that MYR in MYR-PVP in 1:1, 1:2, …, 1:6 and 1:7 weight ratios dissolved in ethanol during the solvent evaporation process precipitation. This effect was not observed for 1:8 and 1:9 weight ratios.

The combination of solvent evaporation and freeze-drying techniques obtained ASDs of MYR-PVP in 1:8 and 1:9 weight ratios. To acquire all the ASDs, the following procedure was implemented: 33.3 mg/30.0 mg (1:8 *w*/*w* and 1:9 *w*/*w* ASDs, respectively) of MYR and 40 mL of ethanol were added to a round-bottom flask (volume 100 mL) and stirred to a visibly clear solution. Then, an accurately weighted amount of PVP30 (266.7 mg and 270 mg for 1:8 *w*/*w* and 1:9 *w*/*w* ASDs) was added and stirred to a visibly clear solution. After that, the flask with the mixture was placed in a BÜCHI B-490 rotary evaporator (Buchi, Switzerland) to remove ethanol under reduced pressure. The water bath was heated up to 40 °C. The process took enough time to visually dry the content of the flask plus 20 min extra to make sure all solvent evaporated. The obtained ASDs were rinsed five times with 10 mL of distilled water to obtain an aqueous solution of MYR-PVP, which was poured onto the steel tray of the shelf freeze dryer. Lyophilization was performed using a Heto PowerDry PL3000 freeze dryer (Thermo Scientific in Waltham, MA, USA). The prepared ASDs were stored in a desiccator at the temperature of 22 °C between studies. Physical mixtures were prepared by weighing an accurate amount of MYR and PVP30 with respect to the weight ratio and mixing two ingredients in a mortar for 10 min.

### 3.3. X-ray Powder Diffraction (XRPD)

On a Bruker D2 Phaser diffractometer (Bruker, Berlin, Germany), diffraction patterns were captured using CuK radiation (1.54060 Å) at tube voltages of 30 kV and tube currents of 10 mA. With a step size of 0.02° 2Θ and a counting rate of 2 s·step^−1^, the angular range was 5° to 40° 2Θ. Origin 2021b software (OriginLab Corporation, Northampton, MA, USA) was used to analyze the obtained data.

### 3.4. Thermogravimetric Analysis (TG)

Thermogravimetric (TG) analysis was performed using TG 209 F3 Tarsus^®^ micro-thermobalance (Netzsch, Selb, Germany). Then, about 10 mg powdered samples were placed in an Al_2_O_3_ 85 µL open crucible and heated at a scanning rate of 10 °C·min^−1^ from 25 °C to 390 °C in a nitrogen atmosphere with a flow rate of 250 mL·min^−1^. The obtained TG data were analyzed using the computer program Proteus 8.0 (Netzsch, Selb, Germany). The visualization of the results was performed using the Origin 2021b software (Origin Lab Corporation, Northampton, MA, USA).

### 3.5. Differential Scanning Calorimetry (DSC)

Thermal analysis was performed using a DSC 214 Polyma differential scanning calorimeter (Netzsch, Selb, Germany). A blank aluminum DSC pan was used as the reference sample. Powdered samples of about 10 mg were placed in sealed pans with a hole in the lid. One heating mode (temperature range 25–400 °C, scanning rate of 10 °C·min^−1^) was used to observe the melting point of MYR in a neat compound. Melting and cooling modes were used to observe the glass transition (T_g_) of MYR-PVP ASDs (↑ 25–180 °C, 10 °C·min^−1^; → 180 °C, 3 min; ↓ 180–25 °C, 40 °C·min^−1^; → 25 °C, 2 min; ↑ 25° C–380 °C, 40 °C·min^−1^, where ↑—melting mode, ↓—cooling mode, →—isothermal mode). A nitrogen atmosphere with a flow rate of 250 mL·min^−1^. The obtained DSC data were analyzed using the Proteus 8.0 software (Netzsch, Selb, Germany). The visualization of the results was performed using the Origin 2021b software (OriginLab Corporation, Northampton, MA, USA).

### 3.6. Fourier-Transform Infrared Spectroscopy (FT-IR), Density Functional Theory (DFT) Calculations, and Molecular Docking

FT-IR spectra of MYR-PVP ASDs were obtained on an IRTracer-100 spectrometer in the range from 4000 cm^−1^ to 400 cm^−1^ at a resolution of 4 cm^−1^. The samples were placed on the ATR diamond crystal and clamped. The ATR clamp head’s precise and even coverage of the crystal with a thin layer of the sample was made possible by the close fit of the head shape to the space above the crystal. A total of 100 scans were taken for each sample. The spectra of samples were compared with crystalline MYR, PVP30, and MYR-PVP physical mixture. With LabSolution IR software (version 1.86 SP2, Shimadzu, Kyoto, Japan), all infrared spectra were collected and then processed, including baseline correction and normalization.

The Gaussian 09 software suite (Wallingford, CT, USA) was used to perform quantum chemical calculations [102]. Using Becke’s three-parameter hybrid functional (B3LYP) density functional theory (DFT) and the 6-311G(d,p) basis set, equilibrium geometries and harmonic frequencies of MYR were computed. The scale factor for the vibrational computations was 0.967. The Origin 2021b program (OriginLab Corporation, Northampton, MA, USA) was used to analyze the acquired data.

A simple MYR-PVP molecular model was prepared according to the protocol published in our previous work [50]. The 3D structure of MYR was obtained from the PubChem database in sdf format (PubChem CID: 5281672; website: https://pubchem.ncbi.nlm.nih.gov/, accessed on 25 August 2023). The initial molecular monomer structure of PVP 30 was prepared using the GaussView software (version 6.0, Wallingford, CT, USA) at B3LYP/6-31G(d,p) basis set of DFT. File conversions were performed using OpenBabel 3.1.1 [103]. To generate amorphous molecular structures of PVP 30, the open-source molecular modeling software Xenoview (version 3.7.9.0, website: www.vemmer.org/xenoview/xenoview.html, accessed on 25 August 2023) was utilized [104]. The amorphous cell was constructed using the Xenoview amorphous builder, adjusting rotatable torsions through a rotational isomeric state (RIS) method. Molecular docking was performed by MGLTools 1.5.6 with AutoDock 4.2 (ADT; Scripps Research Institute, La Jolla, San Diego, CA, USA) [105].

### 3.7. Chromatographic Studies of Changes of Myricetin Concentrations

Changes in MYR concentrations were analyzed using HPLC with the diode array detector method (Shimadzu Corp., Kyoto, Japan). A stationary phase ReproSil-Pur Basic-C18 column (100 × 4.60 mm; 5 µm particle size) was used in a validated procedure to determine concentrations of MYR. 0.1% formic acid and acetonitrile (60:40 *v*/*v*) made up the mobile phase. The column temperature was 30 °C, and the mobile phase flow rate was fixed at 0.55 mL/min. The detecting wavelength was set at 372 nm. LabSolutions LC software (version 6.1.15, Shimadzu Corp., Kyoto, Japan) was used to gather and analyze the data.

### 3.8. Apparent Solubility

Excess amounts of MYR and the ASDs were placed in 5 mL Eppendorf tubes, and 5.0 mL of distilled water was added. All samples were mixed using a vortex mixer for 60 s and were centrifuged in a centrifuge machine HERMILE Z 216 MK (HERMLE Labortechnik GmbH, Wehingen, Germany) (5000 relative centrifugal force (RCF) for 5 min at room temperature). The obtained solutions were filtered through a 0.45 μm nylon membrane filter (Sigma-Aldrich, St. Louis, MO, USA) and analyzed for MYR content using the developed and validated HPLC method. The analysis was performed in triplicate.

The remaining solution after the apparent solubility studies (only for the best ASD) were dried under vacuum at 40 °C and examined for phase transformation by XRPD.

### 3.9. Dissolution Studies

The change in dissolution rate of the best MYR-PVP ASD (based on apparent solubility study) was examined by determining the dissolution rate profiles following the requirements of the European Pharmacopoeia at 37 ± 0.5 °C, using a standard paddle apparatus (Agilent, Santa Clara, CA, USA) with a paddle rotation speed of 50 rpm. The samples containing the MYR-PVP 1:9 *w*/*w* ASD in an amount equivalent to 9.05 mg of MYR were weighed into gelatine capsules and then placed in the spring in order to prevent flotation of the capsule on the surface of the liquid. As dissolution media (volume 500.0 mL), 0.1 mol·L^−1^ hydrochloric acid (pH ~ 1.2) and phosphate buffer (pH ~ 6.8) were used in the pH range corresponding to the gastrointestinal tract environment. At predetermined time intervals (1, 5, 10, 15, 30, 60, 90, 120, 180, 240, 300, 360 min), 2 mL of the sample with the replacement of a pre-warmed dissolution medium was withdrawn. The samples were filtered through 0.22 μm nylon membrane syringe filters. The amount of dissolved MYR was analyzed using the developed and validated HPLC method.

### 3.10. Antioxidant Properties

ABTS, DPPH, CUPRAC, and FRAP were used to determine the antioxidant activity of the water solution of MYR, MYR-PVP 1:8 *w*/*w*, and MYR-PVP 1:9 *w*/*w* samples. All procedures were described previously [46,50,68]. All tests were done in Multiskan GO UV reader (Thermo-Scientific, Waltham, MA, USA).

### 3.11. Anticholinesterase Activity and Molecular Docking Study

A test based on the spectrometric method by Ellman et al. was employed to inhibit both AChE and BChE [106]. This method utilizes thiocholine as an artificial substrate. Through enzymatic interactions with 5,5′-dithio-bis-(2-nitrobenzoic) acid (DTNB), thiocholine is released, leading to the formation of the 3-carboxy-4-nitrothiolate anion (TNB anion). The augmentation in the color intensity of thiocholine on a 96-well plate serves as a spectrophotometric indicator of enzyme activity. Each well contained 25.0 µL of the test sample (water solution of MYR and MYR-PVP obtained in apparent solubility study), 30.0 µL of AChE/BchE solution at a concentration of 0.2 U/mL, and 40.0 µL of 0.05 M Tris-HCl buffer with a pH of 8.0. These were incubated at room temperature for five minutes with agitation. Subsequently, the well was supplemented with 125.0 µL of 0.3 mM DTNB solution and 30.0 µL of 1.5 mM acetylthiocholine iodide (ATCI)/butyrylthiocholine iodide (BTCI) solution, followed by a 20-min incubation under the same conditions. The control sample contained 25.0 µL of distilled water instead of the tested sample. The blank for control was prepared without AchE/BchE replacing it with a TRIS-HCl buffer. The blanks for samples were prepared without AchE/BchE replacing it with TRIS-HCl buffer. The percentage of enzyme inhibition was calculated using the equation [107]:(1)$$\mathrm{AChE}/\mathrm{BChE}\;\mathrm{inhibition}\;(\%)=\frac{1-\left(A_{1}-A_{1b}\right)}{\left(A_{0}-A_{0b}\right)}\cdot 100\%$$
where:

A_1_ is the absorbance of the test sample; A_1b_ is the absorbance of the blank of the test sample; A_0_ is the absorbance of the control; A_0b_ is the absorbance of the blank of the control.

Molecular docking studies were conducted to explore the binding mode between MYR and AChE, as well as BChE enzymes. The active site predictions for AChE and BChE were accomplished using PrankWeb (https://prankweb.cz/, accessed on 25 August 2023) [108,109,110]. The preparation of cholinesterases and the compound involved in the study utilized MGLTools 1.5.6 with AutoDock 4.2 (ADT; Scripps Research Institute, La Jolla, San Diego, CA, USA) [105] and OpenBabel [103]. AutoDockVina 1.1.2. was employed for the actual molecular docking process. The structural interactions were visualized and analyzed using the Protein-Ligand Interaction Profiler (PLIP server, https://plip-tool.biotec.tu-dresden.de/, accessed on 27 August 2023) [111], along with PyMOL 2.5.1. (DeLano Scientific LLC, Palo Alto, CA, USA) [112].

The molecular structure of MYR was obtained from PubChem (PubChem CID: 5281672; website: https://pubchem.ncbi.nlm.nih.gov/, accessed on 25 August 2023) in sdf format. Prior to molecular docking, the geometries of MYR were optimized using the Gaussview software (version 6.0, Wallingford, CT, USA) at B3LYP/6-31G(d,p) basis set of DFT. X-ray crystal structures (in pdb format) of Human AChE (PDB code: 4BDT with 3.10 Å resolution) and human BChE (PDB code: 4BDS with 2.10 Å resolution) were retrieved from the Protein Data Bank. (https://www.rcsb.org/, accessed on 4 February 2023). The preparation of these receptors for docking studies was carried out using AutoDock Tools.

The process involved the elimination of water molecules and bound ligands, the addition of polar hydrogens and Kollman charges, and the merging of non-polar hydrogens. The distance between the enzymes’ surface area and the MYR molecule was restricted to a maximum radius limit of 0.375 Å. The grid box was centered around the active site pocket as predicted by PrankWeb. As predicted by PrankWeb, the AchE active site contained TYR^68^, ASP^70^, TYR^73^, THR^79^, TRP^82^, ASN^83^, GLY^116^, GLY^117^, GLY^118^, TYR^120^, SER^121^, GLY^122^, LEU^126^, GLU^198^, SER^199^, TRP^282^, LEU^285^, SER^289^, PHE^291^, ARG^292^, PHE^293^, TYR^333^, PHE^334^, TYR^337^, TRP^435^, HIS^443^, GLY^444^, TYR^445^. While the BChE’s active pocket contained: ASP^67^, GLY^75^, SER^76^, TRP^79^, ASN^80^, GLY^112^, GLY^113^, GLY^114^, GLN^116^, THR^117^, GLY^118^, TYR^125^, GLU^194^, SER^195^, TRP^228^, PRO^282^, LEU^283^, SER^284^, VAL^285^, ALA^325^, PHE^326^, TYR^329^, PHE^393^, TRP^425^, HIS^433^, GLY^434^, TYR^435^.

### 3.12. Physical Stability

The best MYR-PVP ASD was stored in 2 mL open Eppendorf tubes under ambient conditions and in different humidity and temperature conditions (30 °C/65% RH and 40 °C/75% RH). Research in different humidity and temperature conditions was conducted in the laboratory incubator CLN 32 (Pol-eko Aparatura, Wodzisław Śląski, Poland). Due to the change in the physical form of samples stored at 30 °C/65% RH and 40 °C/75% RH, further tests were carried out only under ambient conditions for three months.

## 4. Conclusions

As a result of the work carried out obtained amorphous solid dispersion of myricetin (MYR) with PVP 30 (PVP) in a 1:8 and 1:9 (*w*/*w*) ratio. The amorphous state of MYR was confirmed by the XRPD (disappearance of Bragg’s peaks) and DSC (disappearance of melting point and observed glass transition) methods. Hydrogen bonds are indicated as stabilizing the amorphous state of MYR, confirmed by both FT-IR analysis and molecular modeling. The presence of a single glass transition temperature confirmed a uniform molecular-level distribution of the MYR within the PVP matrix. The intermolecular hydrogen interaction caused the change MYR crystalline structure to an amorphous form. The formation of strong intermolecular interactions between MYR and PVP molecules led to the receiving of a homogeneous amorphous solid dispersion.

Amorphization contributes to the improved apparent solubility of MYR compared to pure compound. Improvements in the dissolution profile at pH 1.2 and 6.8 were confirmed, with higher results obtained under intestinal fluid due to the weakly acidic nature of MYR.

Improving apparent solubility positively affected the biological activity (such as antioxidant properties and cholinesterase inhibition). The observed improvements in antioxidant activity, as evaluated through DPPH, ABTS, CUPRAC, and FRAP assays, suggest that the amorphous solid dispersion of MYR exhibits enhanced capacity to scavenge free radicals and counteract oxidative stress. The results of our in vitro analysis provide evidence that MYR-PVP ASDs have AChE and BChE inhibitory properties. Whereby the obtained ASDs were more selective toward BChE.

In addition, ASDs demonstrated stability under ambient conditions until two months and suggest potential for long-term use in pharmaceutical production.

The enhanced water solubility, dissolution rate in gastrointestinal fluids, antioxidant activity, and in vitro neuroprotective effects of MYR observed in this study highlight its potential as a versatile and effective therapeutic agent.

Obtained results are based on theoretical and experimental studies, provide valuable insights and pave the way for future investigations into the development of novel formulations utilizing MYR amorphous solid dispersion for various health-related applications.

## Figures and Tables

**Figure 1 molecules-29-01287-f001:**
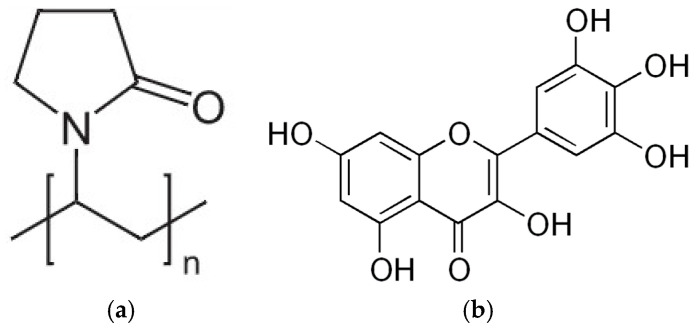
Molecular structures of (**a**) polyvinylpyrrolidone K30 (PVP30) and (**b**) myricetin (3,5,7,3′,4′,5′-hexahydroxyflavone, MYR).

**Figure 2 molecules-29-01287-f002:**
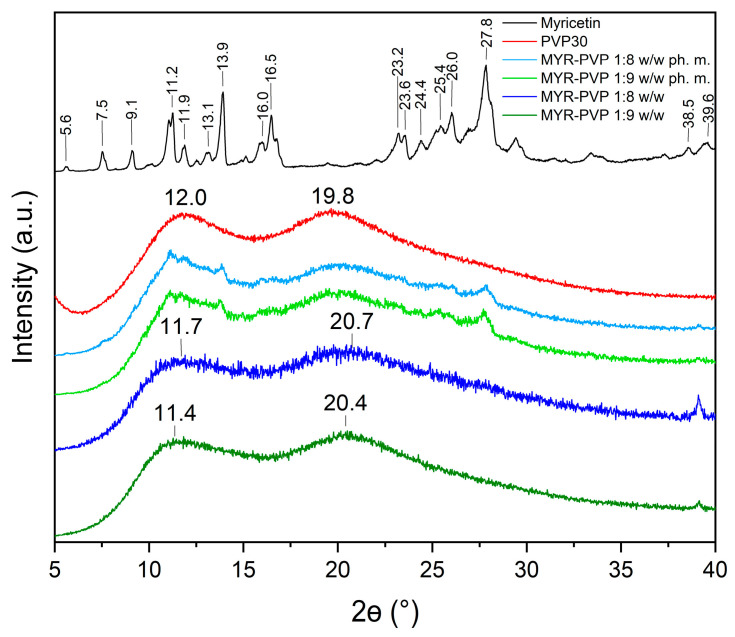
XRPD analysis: myricetin crystalline form (black), PVP30 (red), MYR-PVP 1:8 *w*/*w* ph. m. (light blue), MYR-PVP 1:9 *w*/*w* ph. m. (light green), MYR-PVP 1:8 *w*/*w* ASD (blue), MYR-PVP 1:9 *w*/*w* ASD (green).

**Figure 3 molecules-29-01287-f003:**
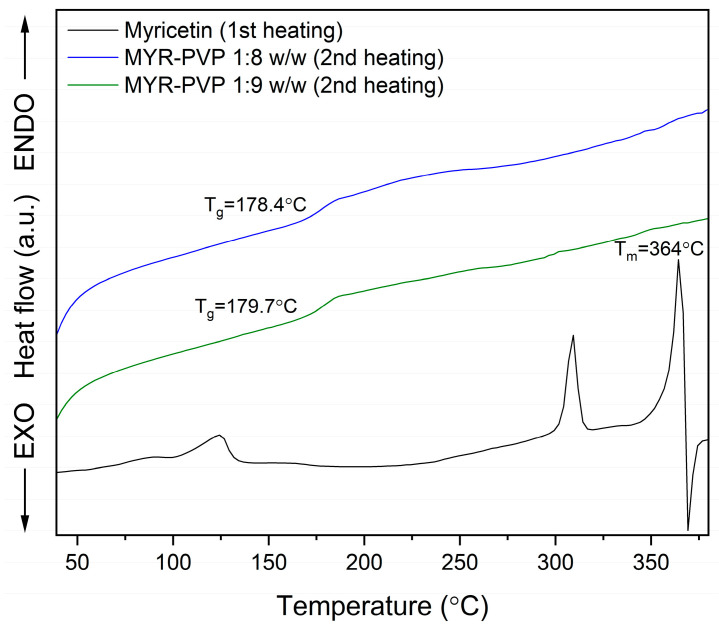
DSC analysis: myricetin (MYR, first heating, black line), myricetin-PVP30 1:8 weight ratio ASD (MYR-PVP 1:8 *w*/*w*, second heating, blue line), myricetin-PVP30 1:9 weight ratio ASD (MYR-PVP 1:9 *w*/*w*, second heating, green line).

**Figure 4 molecules-29-01287-f004:**
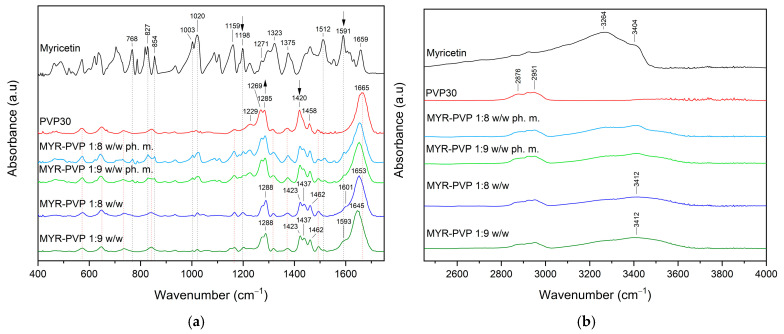
FT-IR analysis: (**a**) Range 400–1750 cm^−1^; (**b**) range 2450–4000 cm^−1^. Legend: myricetin (MYR, black line), PVP30 (red line), myricetin-PVP30 physical mixture 1:8 *w*/*w* (MYR-PVP 1:8 *w*/*w* ph. m., light blue line), myricetin-PVP30 physical mixture 1:9 *w*/*w* (MYR-PVP 1:9 *w*/*w* ph. m., light green line), myricetin-PVP30 ASD 1:8 *w*/*w* (MYR-PVP 1:8 *w*/*w*, blue line), and myricetin-PVP30 ASD 1:9 *w*/*w* (MYR-PVP 1:9 *w*/*w*, green line).

**Figure 5 molecules-29-01287-f005:**
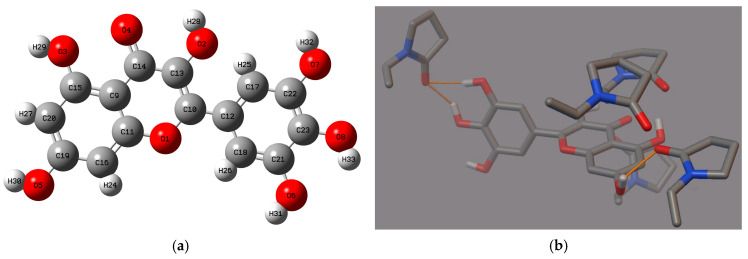
(**a**) The structure of myricetin with the labels of individual atoms, (**b**) myricetin-PVP30 interactions (hydrogen bonds, orange line) of the best poses generated in the Autodock Tool.

**Figure 6 molecules-29-01287-f006:**
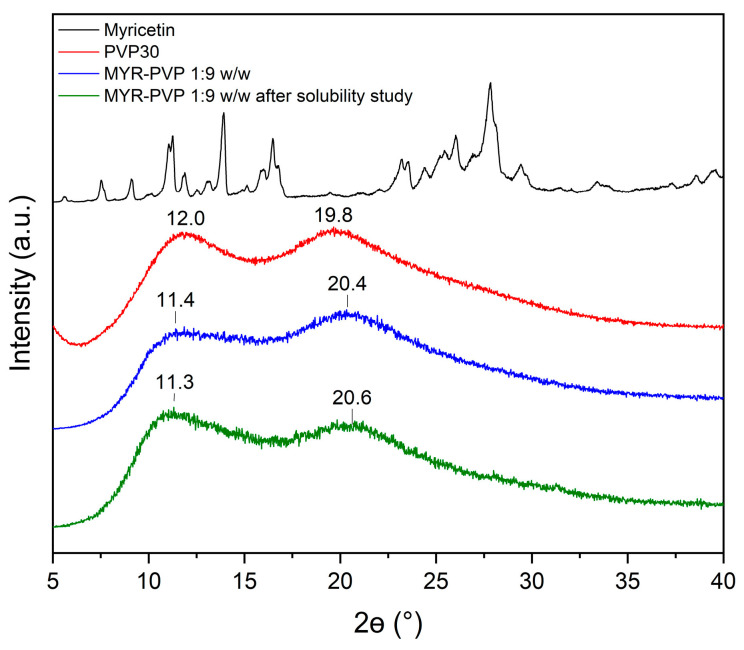
XRPD patterns of MYR (black line), PVP30 (red line), MYR-PVP 1:9 *w*/*w* ASD (blue line), MYR-PVP 1:9 *w*/*w* ASD after the solubility experiment in distilled water (green line).

**Figure 7 molecules-29-01287-f007:**
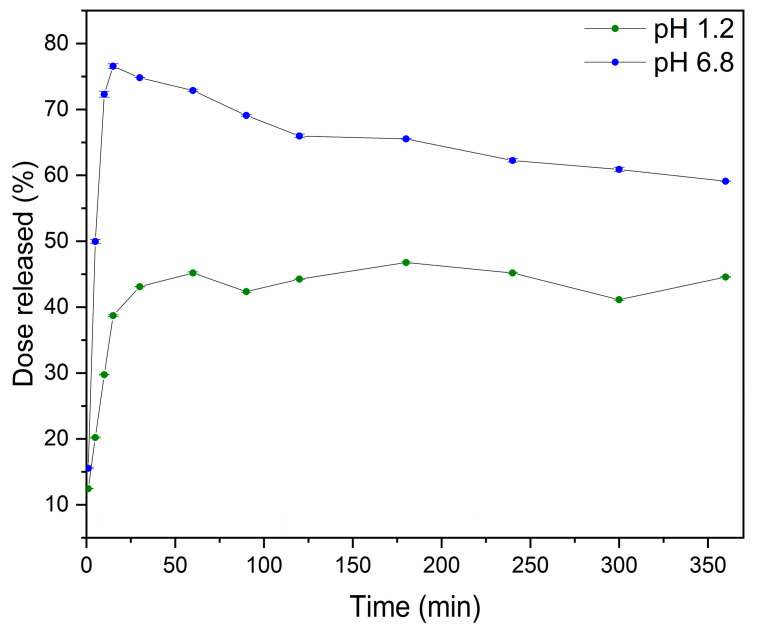
The dissolution rate of MYR in MYR-PVP 1:9 *w*/*w* at pH 1.2 (green) and pH 6.8 (blue).

**Figure 8 molecules-29-01287-f008:**
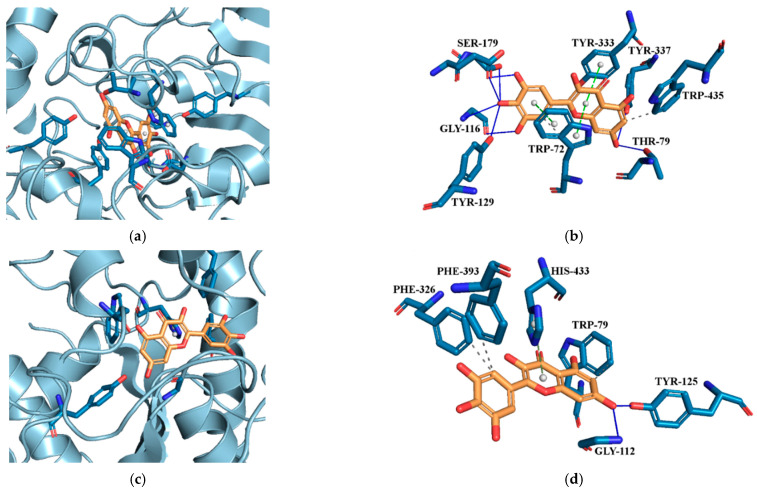
(**a**) Active site gorges of human acetylcholinesterase (AChE, PDB id: 4BDT); (**b**) Proposed binding mode of MYR with AChE; (**c**) Active site gorges of human butyrylcholinesterase (BChE, PDB id: 4BDS); (**d**) Proposed binding mode of MYR with BChE. The key interactions of MYR with residues in the active sites of AChE and BChE are hydrogen bonds (blue solid lines), hydrophobic interactions (grey dashed lines), and π-stacking interactions (green dashed lines). Legend: GLY—glycine, HIS—histidine, PHE—phenylalanine, SER—serine, TRP—tryptophan, TYR—tyrosine.

**Table 1 molecules-29-01287-t001:** The antioxidant activity of the water solution of myricetin and MYR-PVP ASDs.

	ABTS, IC_50_ [µg/mL]	DPPH, IC_50_ [µg/mL]	CUPRAC, IC_0_._5_ [µg/mL]	FRAP, IC_0_._5_ [µg/mL]
Myricetin	none	none	none	none
MYR-PVP 1:8 *w*/*w*	18.30 ± 1.00	54.18 ± 0.95	27.56 ± 0.63	26.74 ± 0.51
MYR-PVP 1:9 *w*/*w*	9.53 ± 0.17	37.52 ± 3.69	20.47 ± 0.29	20.42 ± 0.90

Abbreviation: MYR—myricetin, PVP—polyvinylpyrrolidone, ABTS—2,2′-azino-bis(3-ethylbenzothiazoline-6-sulfonic acid) radical cation-based assays, DPPH—2,2-diphenyl-1-picrylhydrazyl assay, CUPRAC—cupric reducing antioxidant capacity assay, FRAP—ferric reducing ability of plasma assay.

## Data Availability

The data are contained within the article and Appendix A.

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
