# Peer review of "Myricetin Amorphous Solid Dispersions—Antineurodegenerative Potential"

_molecules, 2024, doi:10.3390/molecules29061287_

Round 1

Reviewer 1 Report

Comments and Suggestions for Authors

The article submitted by Natalia Rosiak et al. discussed about myricetin Amorphous solid dispersions and its anti-neurodegenerative potential. The works lacks originality and novelty. The content of the paper is routine work, which cannot attract readers or provide reference for relevant researchers. In my opinion, this manuscript is not suitable for publication in Molecules.

Reviewer 2 Report

Comments and Suggestions for Authors

This study was carefully performed to improve solubility of Myricetin, however, introduction and experimental design should be revised before publication because current from can not present novelty as compared to investigations previously reported. The authors should propose what is novelty of this research; solid dispersion of Myricetin (MYR) was reported before and revised points shown in below. If, the authors would like to insist interaction with proteins, further investigations are required as pharmacological/mechanism research.

1.     Research of the solid dispersion formulations of MYR were reported before. One of them was referred as [54] in this article (page3, line 117-121). The authors discussed the solid dispersion prepared in the article [54] showed the XRD peaks. However, the sample showed the XRD peaks was only in the ratio of MYR-PVP at 80:20. The samples composed of MYR-PVP at 10:90 and 50:50 formed amorphous states.

Could you confirm Fig. 2 in https://doi.org/10.1016/j.molstruc.2017.04.015 ?

2.     Furthermore, a new article reported formation of the solid dispersion of MYR with various polymers. PVP, HPMC, and PEG were used and dissolution profiles of the solid dispersion were compared. This study should be included as reference.

3.     Based on above, the authors should propose a novelty of this investigation, which should be included in introduction. Please revised this article comprehensively (for example, combination of new polymer or evaluation of novel properties such penetration of membrane).

4.     In this study, apparent solubility was designed for distilled water for 60 s. However, simulated gastro-intestinal fluids such as USP-1 and -2 are desired for prediction and solubility. In addition, amorphous formulation often shows precipitation following dissolution profile and dissolution test for hours (https://doi.org/10.1021/acsomega.1c06329) is commonly used for evaluation of solid dispersion. Please re-design the method for solubility/dissolution.

That’s all.

Reviewer 3 Report

Comments and Suggestions for Authors

Dear Sirs,

Notes for authors are included in the text.

In addition, the authors should clarify the following:

Generally, the initial formulations of solid dispersions are prepared with lower proportions of excipients (e.g. active substance/excipient ratios 1:1, 1:2, 1:3, or 1:5). On the basis of which studies did the authors choose MYR: PVP ratios of 1:8 and 1:9 w/w?
